# Relationship between Sleep Bruxism Determined by Non-Instrumental and Instrumental Approaches and Psychometric Variables

**DOI:** 10.3390/ijerph21050543

**Published:** 2024-04-25

**Authors:** Nicole Pascale Walentek, Ralf Schäfer, Nora Bergmann, Michael Franken, Michelle Alicia Ommerborn

**Affiliations:** 1Department of Operative Dentistry, Periodontology, and Endodontology, Faculty of Medicine, Heinrich-Heine-University, Moorenstr. 5, 40225 Düsseldorf, Germany; 2Clinical Institute of Psychosomatic Medicine and Psychotherapy, Faculty of Medicine, Heinrich-Heine-University, Moorenstr. 5, 40225 Düsseldorf, Germany

**Keywords:** sleep bruxism, oral health-related quality of life, anxiety, perceived stress, polysomnography

## Abstract

Sleep bruxism (SB) can be determined with different diagnostic procedures. The relationship between psychometric variables and SB varies depending on the diagnostic method. The aim of the study was to compare the association between SB and oral health-related quality of life (OHRQoL; measured by the Oral Health Impact Profile, OHIP), anxiety (measured by the State-Trait anxiety inventory, STAI), and stress (single scale variable) depending on the diagnostic method in the same sample. N = 45 participants were examined by non-instrumental (possible/probable SB) and instrumental methods (definite SB). The OHIP differed significantly between possible SB (median = 4) and non-SB (median = 0) with W = 115, *p* = 0.01, and probable SB (median = 6) and non-SB (median = 0) with W = 101, *p* = 0.01). There was no significant difference in the OHIP score between definite SB and non-SB. For the other psychometric variables, the analyses revealed no significant differences between SB and non-SB in all diagnostic procedures. The results suggest that there is a difference between possible/probable and definite SB with respect to the association with OHRQoL. Certain aspects of possible/probable SB might be responsible for the poor OHRQoL, which are not measured in definite SB.

## 1. Introduction

Sleep bruxism is a masticatory muscle activity during sleep that is characterized as rhythmic (phasic) or non-rhythmic (tonic) and is neither a movement disorder nor a sleep disorder in otherwise healthy individuals [1]. The rhythmic muscle activity is sometimes accompanied by audible grinding sounds [1,2,3]. In addition to SB, awake bruxism (AB) may also be present, which primarily involves clenching the teeth while awake [4]. Regarding the prevalence of SB, studies reported a rate of 1–15% [5,6].

Depending on the diagnostic method, SB can be classified as possible, probable, or definite SB [1]. Possible SB is based on a positive self-report only. Here, specific criteria are recorded via questionnaires, protocols, or verbal reports [7,8,9]. Probable SB is based on a positive clinical inspection, with or without a positive self-report, and is determined via specific standardized protocols [10,11]. Definite SB is based on a positive instrumental assessment, with or without a positive self-report and/or a positive clinical inspection.

The use of polysomnography (PSG) with audio-video recording is currently referred to as the gold standard diagnostic technique in the determination of definite SB [12,13]. Other instrumental procedures include the use of electromyography, as well as diagnostic sheets [14,15,16]. The validity of questionnaires and clinical examinations is considered to be lower than that of portable devices that measure SB instrumentally [17].

The etiology of SB is multifactorial [1]. Studies could demonstrate that certain genetic predispositions increase the likelihood of the occurrence of SB [18]. Additionally, serotonin seems to have an influence on the presence of SB [19]. Exogenous factors such as alcohol, tobacco, and caffeine also increase the probability of occurrence [20]. Diseases such as Parkinson’s disease and sleep apnea syndrome have also been linked to the presence of SB [21,22]. Regarding consequences, both negative and positive consequences of SB are reported. Regarding negative consequences, it is debated whether SB has an impact on the development of painful temporomandibular dysfunction (TMD) [23,24]. However, other studies report that the increased muscle activity associated with SB is correlated with a reduction of symptoms in both obstructive sleep apnea syndrome and gastroesophageal reflux [25,26]. The focus of the present study is the relationship between psychological factors and SB since these factors are discussed to play a role in the etiology of SB [1]. However, the choice of the underlying diagnostic method must be considered, as findings vary depending on the method used [27]. Studies found a positive association between possible SB and stress [28,29,30], poorer OHRQoL [29], and higher cortisol secretion as an objective marker of stress [31]. Furthermore, a positive correlation was also found between probable SB and psychological stress factors such as anxiety [32,33], stress [34], and cortisol [35,36]. The findings are different for definite SB: Here, studies found an absence of a significant relationship between definite SB and anxiety [37], stress [38], and depression [39]. However, Câmara-Souza et al. reported that definite SB is correlated with worse OHRQoL [40].

The aim of the study was to investigate the relationship between various psychological parameters and SB diagnosis according to the accepted methods. The assumptions examined were that (1) there is a positive correlation between possible/probable (non-instrumental assessed) SB and psychological parameters, and (2) there is no correlation between definite (instrumental assessed) SB and psychological parameters. Psychological parameters included anxiety, stress, and OHRQoL, which were assessed by self-rating questionnaires.

## 2. Materials and Methods

The case-control study was conducted from May 2019 to July 2020. The data come from a monocentric validation study, and the present study is an extension of the analysis of secondary psychometric data based on a previous study [15,41]. The inclusion and exclusion criteria of the sample and the calculation of the sample size are based on the monocentric validation study [15].

A total of four different diagnostic procedures were applied in the same sample: (1) self-report, (2) clinical examination according to AASM criteria (third edition, revised text; ICSD-3 TR), (3) PSG, and (4) diagnostic sheet. Every diagnostic procedure is described in more detail below. Thus, each subject was examined by each method and received the diagnoses of possible SB/non-SB, probable SB/non-SB, and definite SB/non-SB. The psychometric parameters were assessed using self-report tools.

The sample size calculation was based on the original study project [15]. The project aimed to validate a novel method, and accordingly, the sample size planning was pre-established. A power analysis aligned with the specific research question indicated a required sample size of N = 42. To account for potential dropouts, a conservative estimate of a 20% dropout rate was considered. Consequently, the targeted sample size for the study was set at N = 50 subjects (n = 25 subjects with and n = 25 subjects without SB). This approach aimed to ensure the robustness and reliability of the results in light of potential participant attrition during the course of the research.

Subjects underwent four visits and a prior screening, which included different examinations and psychometric assessments. Screening included questionnaires to verify initial inclusion and exclusion criteria and study information. Visit 1 included an educational interview and clinical examination. At visits 2 and 3, a polysomnographic examination with audio-video recording in the subjects’ home environment was performed. At visit 4, subjects were given a newly developed diagnostic sheet that they wore for five consecutive nights to record current SB activity [15].

### 2.1. Subjects

Subjects were recruited via announcements at the University Hospital Düsseldorf, Heinrich Heine University, University of Applied Sciences Düsseldorf, as well as via the institute’s own website and social networks. Eligible subjects had to be physically and mentally healthy and between 20 and 50 years old. General exclusion criteria that would preclude a healthy general condition were as follows: severe mental disorders, central nervous system and/or peripheral nervous system disorders, drug or medication abuse or dependence, and other serious physical or systemic diseases such as autoimmune disease, cardiovascular disease, respiratory insufficiency, active inflammation or malignancy as determined by a medical history questionnaire. Moreover, pregnant and lactating women were excluded from participation. One proficient dentist, as detailed in a previously published source [15], assessed dental inclusion and exclusion criteria through a comprehensive clinical interview and examination. This evaluation also involved confirming signs and symptoms of Temporomandibular Disorders (TMDs) based on the German version of the Research Diagnostic Criteria for TMDs [42]. The presence of TMD symptoms not necessitating treatment was considered a covariate and not an exclusion criterion, as the primary focus of this study was on SB.

### 2.2. Sleep Bruxism Assessment

Possible SB was determined by the question “Do you grind your teeth during sleep?” in a general self-evaluation clinical history protocol. Probable SB was detected when the criteria of the third version of the International Classification of Sleep Disorders (ICSD-3-TR) complied [43]. For this, one trained and experienced dentist systematically examined the subjects’ teeth and conducted an anamnestic interview [10]. Additionally, masseter hypertrophy was assessed according to the criteria of ICSD-2 [44].

Definite SB was determined via PSG with audio-video recording. Measurements were performed on two consecutive nights. Sleep was evaluated according to the AASM criteria [45]. SB activity was assessed according to the diagnostic research criteria of Lavigne et al. [12]. Different patterns or episodes of SB activity are determined. A phasic episode is characterized by at least three muscle contractions (bursts) lasting from 0.25 s to 2.00 s, visibly separated by episodes without activity. A tonic episode is characterized by a sustained muscle contraction of at least 2.00 s duration. An episode of inactivity of at least 3.00 s duration must occur between two episodes for them to be considered separate episodes. A mixed episode is present if this episode of inactivity is absent between a phasic and a tonic episode. A muscle contraction is scored when it exceeds 20% of the individual average amplitude of masseter muscle activity at maximal voluntary clenching (MVC) [12]. Definitive SB is defined when the following characteristics are met: Number of SB episodes/sleep hour > 4, Number of bursts/sleep hour > 25, Number of SB episodes with teeth grinding sound > 1 (minimum 2).

Furthermore, the current SB activity was measured by means of a novel diagnostic sheet including fully automated evaluation software [15]. Subjects wore the 0.5 mm thin sheet for five consecutive nights, and it consisted of five different colored layers. The grinding activity removes material from the sheet and exposes a colored pattern. The sheets were scanned and evaluated using a specific algorithm. The strength of the SB activity is indicated by the so-called pixel score, which combines the area and depth attrited by SB on the sheet. The higher the pixel score, the stronger the current SB activity.

### 2.3. Assessment of Psychometric Parameters

Anxiety was assessed with the State-Trait-Anxiety-Inventory, which contains two scales for the assessment of anxiety as a state and as a trait [46,47]. The questionnaire is a self-report instrument based on 20 questions per scale. Example statements related to the anxiety state are “I am tense” (related to anxiety) or “I am relaxed” (related to freedom from anxiety). The sum values of the respective scales can range from 20 to 80. Higher values represent a stronger intensity of a feeling state, such as inner restlessness or more intense general anxiety. The OHRQoL was recorded with the OHIP-G-14, which consists of 14 statements [48,49]. The statements refer to the period of the past month and thematically focus on difficulties and problems in the area of the mouth, teeth, jaw, or dentures. Disorders in, for example, speech production or food intake are examined. An example item is “In the past month, have you had difficulty pronouncing certain words because of problems with your teeth, mouth, or dentures?” The raw scores are combined into a summed score, which can range from 0 to 56. High scores indicate poor oral health-related quality of life. Stress was assessed with a single Likert scale in the above-mentioned general clinical history. The subjects answered the question, “Is your current life situation burdened by stress?”. A value between 0 (not at all) and 10 (very much) could be given.

The statistical analyses involved descriptive methods for sample characterization, comprising mean values (M) and standard deviations (SD). Frequency data were presented in absolute numbers and relative frequencies as percentages. Group differences were assessed using tests (e.g., Student’s *t*-test), considering scale level and distribution to account for covariates like age, gender, and education. Correlation analyses were conducted using appropriate methods (Pearson, Spearman, biserial) based on variable characteristics. Cohen’s criteria (1988) [50] were applied for interpreting correlation coefficients: 0.1 to 0.3 indicated a weak correlation, 0.3 to 0.5 a medium correlation, and >0.5 a strong correlation. Before statistical analysis, each test’s prerequisite was verified through Shapiro–Wilk tests and visual inspections of quantile–quantile plots for normal distribution. R programming language and R Studio software (Version 4.2.1.; RStudio, Boston, MA, USA) were used for all calculations. To reduce the accumulation of alpha errors in multiple testing, the false discovery rate (FDR) was controlled [51], maintaining a significance level of *p* = 0.05 for all calculations.

## 3. Results

### 3.1. Demographic Characteristics

Forty-five participants were recruited for the study, with n = 22 (48.89%) female and n = 23 (51.11%) male participants. Ages ranged from 21 to 46 years, with M = 26.40 (SD = 4.44). The proportion of students was 78%. German was given as the native language by 93% of the test subjects (others: German/Turkish, n = 1; Russian/German, n = 1; Ukrainian, n = 1). There were no significant differences between the SB and non-SB groups with regard to possible confounding variables (e.g., gender). Data are presented elsewhere [15]. All participants were classified as SB or non-SB according to four different diagnostic procedures. The distribution of the frequency of SB and non-SB depending on the diagnostic procedure is shown in Table 1.

### 3.2. Statistical Group Comparisons of Psychometric Variables

In the first step, the psychometric variables were tested for group differences between SB and non-SB, depending on the diagnostic procedure. Table 2 presents the descriptive statistics of the psychometric secondary variables split by SB vs. non-SB depending on the diagnostic procedure.

The scores of all questionnaires and their subscales are not normally distributed, which is why group comparisons were calculated with the non-parametric Mann–Whitney U-test. The results show that the sum score of OHIP after controlling for FDR differs significantly between subjects with SB (median = 4) and without SB (median = 0) according to self-report (W = 115, *p* = 0.01, power = 0.98). Also, OHIP between SB (median = 6) and non-SB (median = 0) determined by clinical examination differed significantly from each other (W = 101, *p* = 0.01, power = 0.95). All other statistical comparisons were not significant. This means that subjects with SB reported significantly worse oral health-related quality of life after self-reporting and clinical examination. This is not true for SB determined by other methods.

### 3.3. Correlation Analyses of Psychometric Variables and SB Parameter

In the second step, biserial and Spearman correlations were calculated for a more in-depth analysis of the specific diagnostic parameters and the psychometric secondary variables (Table 3). Spearman correlation coefficients were statistically tested. The biserial correlation coefficients could not be statistically tested because the distribution of all secondary psychometric variables was not normally distributed. None of the Spearman correlation coefficients were statistically significant after controlling for FDR.

The number of attritions and the STAI (State and Trait) correlate together with medium strength. This suggests that a higher number of damaged teeth is related to increased anxiety. It is interesting to note that the instrumentally measured parameters (e.g., SB Index and pixel score) show largely weak to no correlations with the psychometric variables. It is possible that instrumentally measured aspects of SB (in this case, current masticatory muscle activity and severity of teeth grinding) are not related to psychological distress. The correlation between SB diagnosis according to PSG (r_b_ = 0.41), as well as SB index (r_s_ = 0.38) with positive stress processing mechanisms, represent an exception. This result suggests that increased masseter muscle activity and the use of positive stress coping mechanisms are related. Consistent with the results from the statistical group comparison, SB (e.g., self-report in the anamnestic interview, r_b_ = 0.66, power = 0.99) and various clinical parameters are strongly related to OHIP. Based on these findings, the OHIP data were examined in more detail.

### 3.4. In-Depth Analysis of the Distribution of OHIP Scores

To better classify the striking result in OHIP, the distribution of OHIP scores is shown and described graphically below. Figure 1 shows the distribution as a histogram. The right-skewed distribution and an outlier at the sum value of 20 are evident.

In the next step, boxplots were created to visually analyze the distribution of OHIP based on the diagnosis and procedure (Figure 2). Here, the discrepancy between SB and non-SB after self-reporting and clinical examination in OHIP becomes apparent. This means that oral OHRQoL is worse in SB after self-report and clinical examination than in non-SB. The discrepancy in OHIP is less pronounced between SB and non-SB diagnosed by instrumental methods.

## 4. Discussion

The results of the present work reveal that subjects with possible (self-report) and probable SB (interview and clinical examination) showed significantly worse OHRQoL than subjects without possible/probable SB. Mean scores were nearly five times higher in SB in each case. Strong positive correlations were present between the SB parameters of the clinical examination and the OHIP-G-14 sum score. This finding is consistent with other studies, which describe a positive association between possible SB and a worse OHRQoL [29,51,52]. To the authors’ knowledge, this is the first study that has examined the association between probable SB and OHRQoL, so no comparisons can be made with other studies. The association was absent when SB parameters of instrumental methods (PSG, diagnostic sheet) were examined. There were no significant group differences between SB and non-SB by PSG or diagnostic sheet, nor did specific SB parameters of these methods correlate with the OHIP-G-14. This suggests that instrumentally recorded SB activity is not reflective of poorer OHRQoL. This finding contradicts that of Câmara-Souza et al., who found a positive correlation between definite SB and OHRQoL [40].

For all other psychometric variables, there were no significant correlations or group differences between SB-specific parameters and the psychometric variables. In particular, the lack of correlation between possible SB and subjective stress in the present study contradicts the results of other studies. This could be reasoned by a different sample size of N = 1784 in Ahlberg et al. or N = 1000 in Emodi Perlman et al. [28,53]. In addition, the sample consisted of adolescents between 12–18 years of age with M ± SD = 15.1 ± 1.5 (thus, on average, 10 years younger than the mean age of the present sample). It is well known that the prevalence of SB decreases with increasing age [5,28,53]. In contrast, the results of Karakoulaki et al. are more difficult to classify because, due to a similar sample composition to the present study (also N = 45), the significantly higher scores in subjective stress experience in SB contradict the present study [31]. Stress in this study was assessed using the Perceived Stress Scale, which provides a more comprehensive picture of subjective stress experience by 10 items. It is possible that capturing stress using a scale, as in the present study is not sufficient to validly measure the impact of stress and its association with SB. However, another study that also applied the Perceived Stress Scale showed no association between subjective stress experience and possible SB but with possible WB [54]. Thus, study results in which stress was measured more comprehensively via a questionnaire are nonetheless inconsistent and require further clarification.

Another frequently studied dimension is anxiety as a state or trait. Here, the present findings contradict those of the study by Kara et al., in which individuals with probable SB showed higher scores on the STAI (both state anxiety and trait anxiety) [32]. Comparability may be limited by the fact that both subjects with and without probable SB had higher mean scores on the STAI than individuals in the present population. In addition, approximately 30% of individuals in the SB group had clinically salient symptoms from the affective disorder spectrum (anxiety disorders). In the present study, individuals with clinically abnormal scores on the GSI were already excluded during screening. To conduct the primary study, which was subject to the guidelines of the German Medical Devices Act, only healthy subjects were allowed to participate in the study. Accordingly, all subjects with conspicuously high psychological distress were excluded. Consequently, the systematic exclusion of individuals with clinically relevant psychiatric or psychological symptoms could lead to a floor effect that could explain the general absence of significant differences in psychological distress between SB and non-SB.

Despite the absence of meaningful findings for the variables stress and anxiety in relation to SB, the finding on the OHRQoL is of interest. The fact that the relationship between SB and the OHIP-G-14 exists only when SB is measured by non-instrumental methods raises further research questions. The results provide an indication that the different diagnostic methods may focus on different facets of SB. This could offer a possible explanation for the different strengths of the correlations. It could be speculated whether other factors and phenomena besides SB are included in self-reported SB, both in the questionnaire and in the interview with the dentist. Possible influences could be the presence of TMD and/or AB. Interestingly, a study by Su et al. showed that the presence of AB correlated more strongly with poor OHRQoL than SB and TMD [55]. Incorporating other findings, the authors discuss that AB leads to daytime pain, and this may correlate with poor OHRQoL [56]. However, it is problematic that non-instrumental methods have yet to be established for AB. This is why a more in-depth investigation of the relationship between AB and psychological stress, including poorer OHRQoL, should be continued. The findings may suggest that individuals with SB, AB, or TMD (or a mix of all phenomena) cannot differentiate clearly via measurement with non-instrumental methods. Thus, the presence of general oral symptoms could be reflected in poorer OHRQoL. In contrast, if SB is measured via instrumental methods, the association between SB and poor OHRQoL is lost. It is possible that non-instrumental methods capture psychological factors such as distress more than instrumental methods. Ohlmann et al. also showed that after calculating a logistic regression, myofascial pain is not explained by SB but by somatization [24]. For further studies, it would be helpful to assess somatization or somatosensory amplification in subjects with SB using questionnaires such as the Screening for Somatoform Disorders or the Somatosensory Amplification Scale [57,58].

Some general and methodological limitations should be mentioned. Due to the study-related exclusion criteria, the variance of psychological distress is limited (exclusion of subjects who have a striking GSI in the SCL-90-S). This could possibly be the reason for the absence of statistically meaningful findings for the variables stress and anxiety. Furthermore, there is partly an uneven distribution in the groups (especially between definite SB, n = 10, and non-SB, n = 35), and the overall sample size is small. Nevertheless, the high test power indicates that the significant differences in OHRQoL between possible/probable SB and non-SB are meaningful. Since this study is a secondary project of another study, the sample size calculation was based on this sample [15]. For future studies, the inclusion of a larger sample with a more balanced distribution between definite SB and non-SB would be important. In addition, a limitation is that stress was only measured with one scale. More detailed questionnaires or a measurement of stress via instrumental methods (cortisol measurement in saliva) would provide more valid findings.

## 5. Conclusions

The strength of the association between psychometric variables varies depending on the diagnostic method. In the present study, SB measured by instrumental methods (PSG, diagnostic slide) was shown to be poorly related to scores on the OHIP-G-14. However, when SB is determined by non-instrumental methods (self-report, clinical examination), subjects with SB show markedly worse OHRQoL. For all other psychometric variables, there were no significant correlations or group differences between SB-specific parameters and the psychometric variables. The result provides evidence that non-instrumental methods may consider the influence of psychological factors, especially focusing on OHRQoL. Crucial factors could be the presence of AB, TMD, or somatization tendencies; therefore, these should be given more attention in the assessment of SB.

## Figures and Tables

**Figure 1 ijerph-21-00543-f001:**
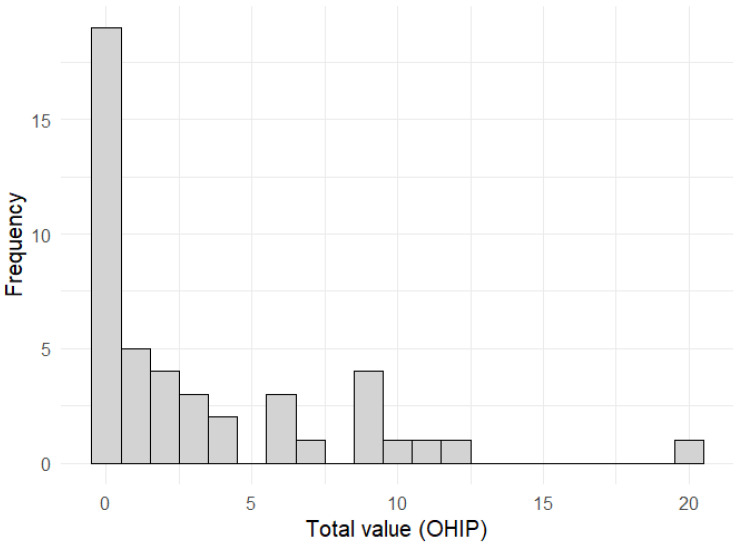
Histogram of the sum values of the OHIP.

**Figure 2 ijerph-21-00543-f002:**
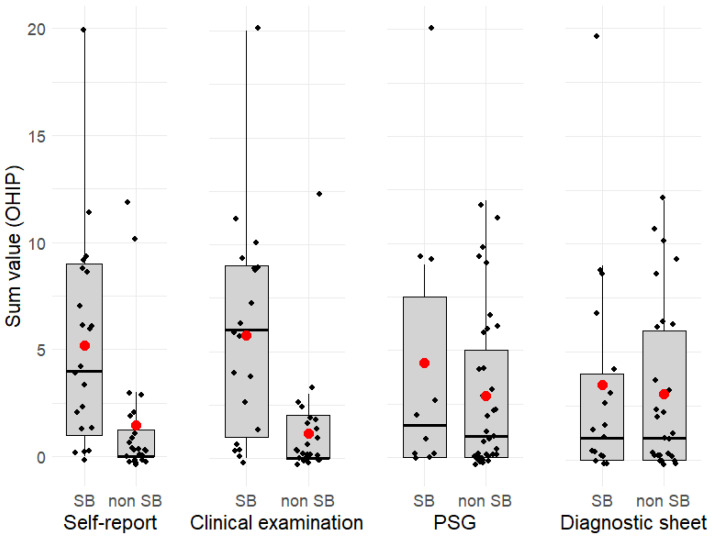
Boxplots of OHIP data divided by diagnosis and procedure. The red dot indicates the mean value. The black dots indicate individual data values.

**Table 1 ijerph-21-00543-t001:** Distribution of SB and non-SB across diagnostic procedures (N = 45).

Diagnostic Procedure	SB n (%)	non-SB n (%)
Self-report	21 (46.67%)	24 (53.33%)
Clinical examination	20 (44.44%)	25 (55.56%)
PSG	10 (22.22%)	35 (77.78%)
Diagnostic sheet	17 (37.78%)	28 (62.22%)

**Table 2 ijerph-21-00543-t002:** Descriptive statistics of psychometric secondary variables split by SB diagnosis per procedure.

	STAI State	STAI Trait	OHIP	Stress
Diagnosis	M ± SD (Median)	M ± SD (Median)	M ± SD (Median)	M ± SD (Median)
Self-report				
SB (n = 21)	34.76 ± 7.89 (34)	38.19 ± 5.59 (37)	**5.19 ± 4.94 (4)**	4.04 ± 2.51 (5)
non-SB (n = 24)	35.33 ± 9.89 (33)	38.71 ± 7.59 (38)	**1.46 ± 3.11 (0)**	4.95 ± 2.11 (4)
Clinical examination				
SB (n = 20)	34.75 ± 8.03 (34)	37.70 ± 5.50 (37)	**5.75 ± 5.05 (6)**	4.95 ± 2.09 (5)
non-SB (n = 25)	35.32 ± 9.72 (34)	39.08 ± 7.52 (38)	**1.16 ± 2.48 (0)**	4.08 ± 2.52 (4)
PSG				
SB (n = 10)	36.80 ± 10.15 (33.5)	38.80 ± 7.81 (36.5)	4.40 ± 6.52 (1.5)	4.50 ± 2.88 (4.5)
non-SB (n = 35)	34.57 ± 8.63 (34)	38.37 ± 6.42 (38)	2.86 ± 3.70 (1)	4.46 ± 2.23 (4)
Diagnostic sheet				
SB (n = 17)	36.06 ± 8.19 (34)	38.88 ± 6.77 (37)	3.47 ± 5.28 (1)	4.25 ± 2.08 (5)
non-SB (n = 28)	34.46 ± 9.43 (34)	38.21 ± 6.71 (38)	3.04 ± 3.93 (1)	4.82 ± 2.77 (4)

Note: Values in bold indicate significant group differences in the non-parametric Mann–Whitney U-test (SB vs. non-SB).

**Table 3 ijerph-21-00543-t003:** Correlation coefficients of the diagnostic SB parameters and the psychometric variables.

SB Parameters	STAI State	STAI Trait	OHIP	Stress
Self-report				
Diagnosis SB (y/n) ^‡^	−0.04	−0.05	0.53	0.25
Clinical examination				
Diagnosis SB (y/n) ^‡^	−0.04	−0.13	0.65	0.23
SB external-report (y/n) ^‡^	0.23	0.02	0.66	0.24
Masseterhypertrophy (y/n) ^‡^	−0.03	0.20	0.32	0.17
Number of teeth with attrition ^†^	0.34	0.34	0.13	−0.06
PSG				
Diagnosis SB (y/n) ^‡^	0.14	0.04	0.20	0.01
SB index ^†^	0.13	−0.07	0.12	0.08
SB episodes with grinding sounds ^†^	−0.18	−0.09	0.18	0.10
Diagnostic sheet				
Diagnosis SB (y/n) ^‡^	0.11	0.06	0.06	0.15
Pixelscore ^†^	0.02	−0.03	0.10	0.03

^†^ Spearman correlation coefficients. ^‡^ Biserial correlation coefficients.

## Data Availability

The datasets generated during and/or analyzed during the current study are available from the corresponding author upon reasonable request.

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
