# Peer review of "Relationship between Sleep Bruxism Determined by Non-Instrumental and Instrumental Approaches and Psychometric Variables"

_ijerph, 2024, doi:10.3390/ijerph21050543_

Round 1

Reviewer 1 Report

Comments and Suggestions for Authors

Please add about validating the questionnaire used in this study in the method section. ( to measure possible SB , stress, and anxiety quest)

sample size 50 is it for all? Because the design is case-control, should it be 25  cases and 25 controls?--> please clarify 

Are there any confounding variables analyzed in this study such as age, sex or education level  and discussion regarding this 

Author Response

Responds to Reviewer

Thank you very much for taking the time to edit our article. Your comments were helpful and constructive. We have carefully reviewed and discussed all of your comments. Below you will find a list of all your commentaries together with our response. Thank you again for your time and care!

Point-by-point answer

  1. Please add about validating the questionnaire used in this study in the method section. ( to measure possible SB , stress, and anxiety quest)

Authors note: Thank you for your comment. We have used a general clinical history protocol to record possible SB and stress. We have specified this:

Possible SB was determined by the question "Do you grind your teeth during sleep?" in a general self-evaluation clinical history protocol. (p. 3, ll. 125-126).

Stress was assessed with a single Likert scale in the above mentioned general clinical history protocol. (p. 4, ll. 168-169).

With regard to the State-Trait Anxiety Inventory, we refer to the relevant publications of the authors, which also include tests of the validity of the inventory (Spielberger, C.D.; Gonzalez-Reigosa, F.; Martinez-Urrutia, A.; Natalicio, L.F.S.; Natalicio, D.S. The state-trait anxiety inventory. Interam. J. Psychol. 1971, 5; Laux, L.; Glanzmann, P.; Schaffner, P.; Spielberger, C.D. Das state-trait-angstinventar [The state-trait anxiety inventory]. Hogrefe, Göttingen 1981)

  1. sample size 50 is it for all? Because the design is case-control, should it be 25  cases and 25 controls?--> please clarify 

Authors note: Thank you for the important comment. We have clarified the relevant section: 

Consequently, the targeted sample size for the study was set at N = 50 subjects (n = 25 subjects with and n = 25 subjects without SB). (p. 3, ll. 96-97).

  1. Are there any confounding variables analyzed in this study such as age, sex or education level and discussion regarding this 

Authors note: Thank you for your comment. We have already tested confounding variables in the preliminary study, and there are no significant differences between the groups. We have added the relevant passage: 

There were no significant differences between the SB and non-SB groups with regard to possible confounding variables (e.g. gender). Data is presented elsewhere [41]. (p. 4, ll. 192-194).    

Reviewer 2 Report

Comments and Suggestions for Authors

The authors try to find the relationship between 4 methods to determine sleep bruxism with stress (single scale variable), anxiety (STAI) and OHRQoL.

They find very low correlation values, however, these values cannot be valid, since the standard deviations of their data are too high, so it is not possible to analyze statistically with that amount of sample, mainly PSG (n=10).

They themselves mention at the end of the discussion of the article, that it is necessary to increase their sample size, because of their findings after their statistical analysis.

In summary, it is necessary to increase the sample mainly of patients with bruxism (mainly diagnosed with PSG) to really know the correlation with anxiety, stress and OHRQoL.

Author Response

Responds to Reviewer 

Thank you very much for taking the time to edit our article. Your comments were helpful and constructive. We have carefully reviewed and discussed all of your comments. Below you will find a list of all your commentaries together with our response. Thank you again for your time and care!

Point-by-point answer

The authors try to find the relationship between 4 methods to determine sleep bruxism with stress (single scale variable), anxiety (STAI) and OHRQoL.

  1. They find very low correlation values, however, these values cannot be valid, since the standard deviations of their data are too high, so it is not possible to analyze statistically with that amount of sample, mainly PSG (n=10). They themselves mention at the end of the discussion of the article, that it is necessary to increase their sample size, because of their findings after their statistical analysis. In summary, it is necessary to increase the sample mainly of patients with bruxism (mainly diagnosed with PSG) to really know the correlation with anxiety, stress and OHRQoL.

Authors note: Thank you for your detailed comment. We agree with the reviewer that the unequal distribution of SB and non-SB subjects after categorization by PSG complicates the interpretation of the results, which we discuss (please, see p. 9, ll. 338-340). However, we have concretized the passage:

 For future studies, the inclusion of a larger sample with a more balanced distribution between definite SB and non-SB would be important. (p. 9, ll. 342-343)

We would like to point out that not all correlation coefficients are low, but differentiate between SB depending on the diagnosis. Possible SB and oral health-related quality of life, for example, show a high correlation (see table 3, p. 3).     

Reviewer 3 Report

Comments and Suggestions for Authors

The authors are presenting a research  to investigate the relationship between various psychological parameters and SB diagnosed according to the accepted methods.

The study was well conducted and the result correctly presented, but the study samples seems very reduced. In order to consider the article for publication in my opinion is advisable to justify the sample size from a statistical point of view, the power was not stated, and to discuss the results reporting about the sample size of similar studies 

Moreover the sample included non german speaking population, how did you avoid comprension problems during testing?

Comments on the Quality of English Language

Dear Editor 

thanks for relying on my expertise to review this paper

the pertinent comments are enclosed in the suggestions for authors section

kind regards

Author Response

Responds to Reviewer

Thank you very much for taking the time to edit our article. Your comments were helpful and constructive. We have carefully reviewed and discussed all of your comments. Below you will find a list of all your commentaries together with our response. Thank you again for your time and care!

Point-by-point answer

The authors are presenting a research  to investigate the relationship between various psychological parameters and SB diagnosed according to the accepted methods.

  1. The study was well conducted and the result correctly presented, but the study samples seems very reduced. In order to consider the article for publication in my opinion is advisable to justify the sample size from a statistical point of view, the power was not stated, and to discuss the results reporting about the sample size of similar studies.

Authors note: Thank you for your comment. We have calculated and supplemented the power of the statistically significant results post hoc, and the values are high:

The results show that the sum score of OHIP after controlling for FDR differs significantly between subjects with SB (median = 4) and without SB (median = 0) according to self-report (W = 115, p = .01, power = 0.98). Also, OHIP between SB (median = 6) and non-SB (median = 0) determined by clinical examination differed significantly from each other (W = 101, p = .01, power = 0.95). (p.5, ll. 215-219)

 It can therefore be stated that despite the small sample size, the power of the significant results is high and the results are therefore meaningful. We have refrained from specifying the power of the other results due to the amount of data, but would add it if desired. We also refer to the power in the discussion:

Nevertheless, the high test power indicate that the significant differences in OHRQoL between possible/probable SB and non-SB are meaningful. (p. 9, ll. 346-348).

  1. Moreover the sample included non german speaking population, how did you avoid comprension problems during testing?

Authors note: Thank you for this important information. A lack of German language skills is listed as an exclusion criterion, which is based on the exclusion criteria of the previous study ([41] Ommerborn, M. A., Walentek, N., Bergmann, N., Franken, M., Gotter, A., & Schäfer, R. (2022). Validation of a new diagnostic method for quantification of sleep bruxism activity. Clinical Oral Investigations, 26(6), 4351-4359.). Please, see p. 2, ll. – 81-82: Inclusion and exclusion criteria of the sample and the calculation of the sample size are based on the monocentric validation study [41].

Reviewer 4 Report

Comments and Suggestions for Authors

Dear Authors,

I reviewed the manuscript entitled "Relationship between Sleep Bruxism determined by Non-instrumental and Instrumental Approaches and Psychometric Variables" for International Journal of Environmental Research and Public Health. The authors investigated if the relationship between psychometric variables and SB varies depends on the diagnostic method. It’s a really nice manuscript to read, and the study is well-performed. However, I do have many points to discuss, including an aspect of great concern.

I encourage the authors that the manuscript is checked by a native English speaking person or a professional English editing service, because of several grammar mistakes, such as … characterized as rhythmic (phasic) or non-rhythmic (tonic) and is whether [wrong word!] a movement disorder nor a…

In the introduction, the authors write that SB is accompanied by audible crunching sounds reported by the sufferers themselves or relatives/bed partners [2,3]. Is it possible to make clear that the bruxing person can’t notice these sounds because he/she is asleep? In other words, a single living bruxer without bed partners in fact can’t report this behaviour.

I don’t agree with the suggestion that possible SB, which is based on a positive self-report, can only be recorded by means of specific criteria via questionnaires or protocols [7-9]. Any self-report will do, including a patient who just informs the dentist verbally.

I don’t think this sentence is correct: …the relationship between psychological factors and SB, since these factors are discussed to play a major role in the etiology of SB [1]. Most experts agree with the fact that the relationship between SB and psychological variables is still a point of discussion. Unlike awake bruxism, there is still no consensus of a clear relationship. This brings me to the next point, I really miss the information about what is known about relationship between psychological factors and SB. Mention more papers.

To my surprise, the aim of the manuscript under review is not clear to me. It lacks a clear explanation that diagnoses of possible, probable, and definite sleep bruxism are being compared. An alternative would be to mention the words subjective, non-instrumental methods, and instrumental methods instead of this rather vague aim ‘to investigate the relationship between various psychological parameters and SB diagnosed according to the accepted methods’.

A point of greater concern, is that this manuscript contains almost identical information (type of study, number of participants, methods and conclusion) as this recently published paper by the same author [Walentek et al. 2023-Association between psychological distress and possible, probable, and definite sleep bruxism]. The only difference that I see is that one study used the global severity index (GSI) of the Symptom-Checklist-90-S to represent psychological distress, and the other used the Oral Health Impact Profile, anxiety, and stress. I might be wrong, but this smells like some kind of plagiarism… I warn the authors that this manuscript can be retracted in case the editor believes that this is a redundant publication (i.e. when the authors present the same data in several publications).

Comments on the Quality of English Language

See comments 

Author Response

Responds to Reviewer

Thank you very much for taking the time to edit our article in very great detail. Your comments were helpful and constructive. We have carefully reviewed and discussed all of your comments. Below you will find a list of all your commentaries together with our response. Thank you again for your time and care!

Point-by-point answer

Dear Authors,

I reviewed the manuscript entitled "Relationship between Sleep Bruxism determined by Non-instrumental and Instrumental Approaches and Psychometric Variables" for International Journal of Environmental Research and Public Health. The authors investigated if the relationship between psychometric variables and SB varies depends on the diagnostic method. It’s a really nice manuscript to read, and the study is well-performed. However, I do have many points to discuss, including an aspect of great concern.

  1. I encourage the authors that the manuscript is checked by a native English speaking person or a professional English editing service, because of several grammar mistakes, such as … characterized as rhythmic (phasic) or non-rhythmic (tonic) and is whether [wrong word!] a movement disorder nor a…

Authors note: Thank you for pointing this out. The manuscript was checked and grammatical and other errors were corrected.

  1. In the introduction, the authors write that SB is accompanied by audible crunching sounds reported by the sufferers themselves or relatives/bed partners [2,3]. Is it possible to make clear that the bruxing person can’t notice these sounds because he/she is asleep? In other words, a single living bruxer without bed partners in fact can’t report this behaviour.

Authors note: We would like to thank you explicitly for this important information. We agree that the wording is difficult. We have followed the consensus paper by Lobbezoo et al. (2018):

Approaches for assessing sleep bruxism based on self-report, although theoretically more difficult than for awake bruxism as the patient is asleep whilst performing the activity, do allow for more options. Specifically, multiple informants can be interrogated, viz., not only the patients themselves but also their bed partner or—in the case of children—their parents. (Lobbezoo, F., Ahlberg, J., Raphael, K. G., Wetselaar, P., Glaros, A. G., Kato, T., ... & Manfredini, D. (2018). International consensus on the assessment of bruxism: Report of a work in progress. Journal of oral rehabilitation, 45(11), 837-844. https://doi.org/10.1111/joor.12663)

We also agree with the statement that a bruxer living alone without a bed partner may indeed not report this behavior. Due to the pursuit of a different research question, we refrain from a method-critical discussion and refer to the existing criteria in the ICSD-3 TR: A. Presence of regular or frequent audible teeth grinding during sleep.

We have revised the sentence:

The rhythmic muscle activity is sometimes accompanied by audible grinding sounds [1-3]. (p. 1, ll. 32-35).

  1. I don’t agree with the suggestion that possible SB, which is based on a positive self-report, can only be recorded by means of specific criteria via questionnaires or protocols [7-9]. Any self-report will do, including a patient who just informs the dentist verbally.

Authors note: Thank you for your helpful comment. We have adjusted the relevant text passage:

Possible SB is based on a positive self-report only. Here, specific criteria are recorded via questionnaires, protocols or verbal report [7-9]. (p. 1, ll. 39-40)

  1. I don’t think this sentence is correct: …the relationship between psychological factors and SB, since these factors are discussed to play a major role in the etiology of SB [1]. Most experts agree with the fact that the relationship between SB and psychological variables is still a point of discussion. Unlike awake bruxism, there is still no consensus of a clear relationship. This brings me to the next point, I really miss the information about what is known about relationship between psychological factors and SB. Mention more papers.

Authors note: Thank you for your helpful comment. We agree with your point and have adjusted the relevant passage:

The focus in the present study is the relationship between psychological factors and SB, since these factors are discussed to play a role in the etiology of SB [1]. (p. 2, ll. 61-62).

In the following section, we have cited several studies that have already reported a relationship between SB and psychological parameters, so we have refrained from further studies due to the diversity that already exists:

Studies found a positive association between possible SB and stress [28-30], poorer OHRQoL [29], and higher cortisol secretion as an objective marker of stress [31]. Furthermore, a positive correlation was also found between probable SB and psychological stress factors such as anxiety [32,33], stress [34], and cortisol [35,36]. The findings are different for definite SB: Here, studies found an absence of a significant relationship between definite SB and anxiety [37], stress [38], and depression [39]. However, Câmara-Souza et al. reported that definite SB is correlated with worse OHRQoL [40]. (p. 2, ll. 64-71)

  1. To my surprise, the aim of the manuscript under review is not clear to me. It lacks a clear explanation that diagnoses of possible, probable, and definite sleep bruxism are being compared. An alternative would be to mention the words subjective, non-instrumental methods, and instrumental methods instead of this rather vague aim ‘to investigate the relationship between various psychological parameters and SB diagnosed according to the accepted methods’.

Authors note: Thank you for your careful correction. We would like to point out that in the following passage we already refer to specific assumptions of the study that contain the above-mentioned specifications (non-instrumental and instrumental methods):

The aim of the study was to investigate the relationship between various psychological parameters and SB diagnosed according to the accepted methods. The assumptions examined were that (1) there is a positive correlation between possible/probable (non-instrumental assessed) SB and psychological parameters; and (2) there is no correlation between definite (instrumental assessed) SB and psychological parameters. Psychological parameters included anxiety, stress, and OHRQoL, which were assessed by self-rating questionnaires.

  1. A point of greater concern, is that this manuscript contains almost identical information (type of study, number of participants, methods and conclusion) as this recently published paper by the same author [Walentek et al. 2023-Association between psychological distress and possible, probable, and definite sleep bruxism]. The only difference that I see is that one study used the global severity index (GSI) of the Symptom-Checklist-90-S to represent psychological distress, and the other used the Oral Health Impact Profile, anxiety, and stress. I might be wrong, but this smells like some kind of plagiarism… I warn the authors that this manuscript can be retracted in case the editor believes that this is a redundant publication (i.e. when the authors present the same data in several publications).

Authors note: Thank you for this important comment. The present data is part of large investigation. Other partial results have recently been published. We regret our error in forgetting to cite the paper. In our correction, we now refer to the paper:

The data of the present case-control study represent a partial analysis of a monocentric validation study that was conducted from May 2019 to July 2020. Initial results of other scientific questions have already been published [41, 42].

[42] =  https://doi.org/10.3390/jcm13020638

Round 2

Reviewer 3 Report

Comments and Suggestions for Authors

Dear authors

The comments have been adequately addressed

The paper has been sufficiently improved

Yours sincerely

Comments on the Quality of English Language

English is fine

Reviewer 4 Report

Comments and Suggestions for Authors

No more comments. I congratulate the authors with their paper!